# Deciphering Acute Myeloid Leukemia Associated Transcription Factors in Human Primary CD34+ Hematopoietic Stem/Progenitor Cells

**DOI:** 10.3390/cells13010078

**Published:** 2023-12-29

**Authors:** Sophie Kreissig, Roland Windisch, Christian Wichmann

**Affiliations:** Division of Transfusion Medicine, Cell Therapeutics and Haemostaseology, LMU University Hospital, LMU Munich, 81377 Munich, Germany; sophie.kreissig@med.uni-muenchen.de (S.K.);

**Keywords:** acute myeloid leukemia, transcription factors, ex vivo expansion, in vitro cell models

## Abstract

Hemato-oncological diseases account for nearly 10% of all malignancies and can be classified into leukemia, lymphoma, myeloproliferative diseases, and myelodysplastic syndromes. The causes and prognosis of these disease entities are highly variable. Most entities are not permanently controllable and ultimately lead to the patient’s death. At the molecular level, recurrent mutations including chromosomal translocations initiate the transformation from normal stem-/progenitor cells into malignant blasts finally floating the patient’s bone marrow and blood system. In acute myeloid leukemia (AML), the so-called master transcription factors such as RUNX1, KMT2A, and HOX are frequently disrupted by chromosomal translocations, resulting in neomorphic oncogenic fusion genes. Triggering ex vivo expansion of primary human CD34+ stem/progenitor cells represents a distinct characteristic of such chimeric AML transcription factors. Regarding oncogenic mechanisms of AML, most studies focus on murine models. However, due to biological differences between mice and humans, findings are only partly transferable. This review focuses on the genetic manipulation of human CD34+ primary hematopoietic stem/progenitor cells derived from healthy donors to model acute myeloid leukemia cell growth. Analysis of defined single- or multi-hit human cellular AML models will elucidate molecular mechanisms of the development, maintenance, and potential molecular intervention strategies to counteract malignant human AML blast cell growth.

## 1. Introduction: AML and Driver Fusion Genes

Hematological malignancies arise from components of the blood forming system including myeloid and lymphoid compartments, but highly differ in their severity and prognosis. Depending on their location and cells of origin, these diseases can be classified into leukemia, lymphoma, or multiple myeloma. All hematological malignancies arise from uncontrolled proliferation and blocked differentiation of immature progenitor cells, floating the blood system with blast cells interfering with regular blood cell functions [1]. Among the adult population, the most common type of acute leukemia is acute myeloid leukemia (AML), where the disease originates in hematopoietic bone marrow progenitors. AML accounts for approximately 80% of all cases and has the shortest five-year survival rate of all leukemias [2]. Genetic alterations enabling the malignant transformation of the hematopoietic progenitors include over-activated proliferative signaling, blockade of differentiation and apoptosis, metabolic remodeling, drug response and evasion, replicative immortality, and genome instability propelling further gene mutations [3,4,5]. The main causes of AML are chromosomal abnormalities resulting from translocations, point mutations, or copy number alterations. Particularly, chromosomal rearrangements such as deletions, duplications, inversions, and translocations lead to malignant transformation by aberrant expression of oncogenes or tumor-specific fusion transcripts [6]. In AML it has been shown that around 13 somatic mutations must occur for full-blown leukemia, hence it is regarded as a multi-step process mostly affecting patients at older ages [7]. Originally, mutations in AML were classified in oncogenic transcription factors and activated receptor tyrosine kinases, blocking differentiation and propelling proliferation [4]. However, Other genes, such as chromatin modifiers, cohesion complex components, spliceosome genes, and tumor suppressors, are also involved.

Since the first discovery of chromosomal translocation in a patient with chronic myeloid leukemia (CML) in 1972 [8], dozens of other chromosomal translocations and corresponding oncogenic fusion-proteins have been identified among leukemic entities, including AML. Based on the chromosomal abnormalities the World Health Organization (WHO) divides AML into several groups [9], whereas the International Consensus Classification (ICC) divides AML by current diagnostic criteria, genetic abnormalities, and blast count [10]. A summary of defining genetic abnormalities associated with AML according to WHO and ICC classification is shown in Table 1.

## 2. Defined Chimeric Transcription Factors Expand Human CD34+ Progenitors in Ex Vivo Cultures as Single Factors

Starting from chromosomal translocations as initial genetic lesions detected in leukemia patients and thereof derived fusion oncoproteins, several research groups aimed to model oncogenic transformation of myeloid blood progenitors in vitro. To do so, such oncogenic transcription factors were introduced into human CD34+ hematopoietic stem/progenitor cells from healthy donors to study the steps of cell transformation and clonal expansion. Table 2 summarizes the oncogenes used to expand human CD34+ cells ex vivo and indicates culture conditions as well as resulting phenotypes.

The following sections detail how the various oncogenes contribute to the leukemic transformation of human CD34+ blood cell progenitors and how genetic engineering can be used to study leukemic ex vivo progenitor cell proliferation (Figure 1).

### 2.1. KMT2A Rearrangements (KMT2A-r)

*KMT2A* (Mixed Lineage Leukemia, *KMT2A,* and *MLL*) rearrangement (*KMT2A*-r) derived fusion proteins are strong drivers of leukemic transformation. *KMT2A* translocations can cause different forms of hematological malignancies including acute myeloid leukemia, different types of acute lymphocytic leukemia, myelodysplastic syndrome, and lymphoblastic lymphoma and therefore affect more than one specific cell type [21,22]. In acute myeloid leukemia, *KMT2A* translocations are found in about 10% of all cases and with particularly high frequencies in pediatric patients (15–20%) and infants (approximately 50%) [23]. Physiologically, the histone methyl transferase KMT2A is expressed in a broad variety of tissues and consists of several highly conserved protein domains crucial for its function. KMT2A is generally involved in epigenetic regulation of developmental and hematopoiesis-associated target genes. Homozygous knockout of *KMT2A* in mice was shown to be embryonically lethal, whereas heterozygous *KMT2A* deficient embryos displayed severe misdevelopments including growth and skeletal disorders, as well as impaired hematopoiesis [24]. The expression interrelationship between KMT2A and highly conserved homeobox genes is crucial for normal hematopoiesis and development. Studies showed that KMT2A is not necessary for initiation but for maintenance of *HOX* gene expression [25,26]. Direct transcriptional targets of KMT2A include the homeobox gene *HOXA9* and HOX cofactor *MEIS1*, along with a broad network of genes up- or downregulated directly by KMT2A, orchestrating hematopoiesis [27].

Epigenetic activation or repression of associated genes is accomplished by chromatin remodeling through direct binding to promotor regions and methyl transferase activity [28,29]. In the case of chromosomal translocation, the 5′ *N*-terminus of *KMT2A* fuses to the 3′ C-terminus of more than 100 fusion partners, leading to various chimeric functional genes. Even though *KMT2A* can fuse to a variety of genes, over 90% of *KMT2A* rearrangements involve nine specific gene fusions, notably fusions to *AF4* (*AFF1*), *AF9* (*MLLT3*), and *ENL* (*MLLT1*) accounting for 36%, 19%, and 13% of all AML cases with *KMT2A*-, respectively [30].

Even though *KMT2A* rearrangements were studied in vitro and in vivo mainly in syngeneic murine models since the 1990s [31,32], differences between mice and humans should not be underestimated. Especially considering hematopoiesis, mouse models cannot fully reflect the human counterpart [33]. Immunophenotyping of human and murine HSPCs for example showed distinctly different expression of cell surface markers such as CD34 or FLT3, which are only expressed on human HSPCs but not in mice [34]. Further, immunological processes differ between mice and humans, which could distort the comparison of effects [35]. Considering these divergences, research on primary human cells is highly relevant but up to date only little data is available in the human context. Montes and colleagues were one of the first to investigate the oncogenic effects of KMT2A::AF4 in vitro in primary human CD34+ hematopoietic stem/progenitor cells (HSPCs) [15]. AF4 interacts with several protein complexes, such as pTEFb, DOT1L, or SL1, and thus is involved in RNA polymerase II recruitment and phosphorylation, as well as transcriptional elongation [36,37,38]. The *KMT2A::AF4* in-frame fusion gene was cloned together with an EGFP reporter in a lentiviral vector backbone. Ex vivo liquid culture of KMT2A::AF4 transduced human umbilical cord blood HSPC CD34+ cells, notably without the need for additional feeder cells support, revealed phenotypical and functional differences between KMT2A::AF4 and non-transduced hCD34+ cells. Upon overexpression of KMT2A::AF4 enhanced growth, reduced apoptosis and increased clonogenic growth could be observed. Further, KMT2A::AF4 modulated *HOX* gene expression profiles. For instance, enhanced HOXA9 expression was measured in KMT2A::AF4+ progenitor cells. This demonstrated as a proof-of-principle that KMT2A::AF4 can immortalize human primary CD34+ HSPCs in vitro resulting in long-term ex vivo liquid cultures [15]. Besides using retroviral vectors to induce KMT2A::AF4 overexpression in CD34+ HSPCs in vitro, CRISPR-Cas9 gene-editing [39,40,41] was used to introduce the t(4;11) translocation in primary human fetal liver HSPCs (FL HSPCs) isolated from donated fetal liver tissue, to investigate the transforming capacity of KMT2A::AF4 in vitro. To induce the *KMT2A::AF4* translocation, FL HSPCs were electroporated with Cas9/KMT2A- and Cas9/AF4-single guide RNA nucleotides. In accordance with the most common breakpoint regions single guide RNAs were designed to target intron 11 and intron 3 of *KMT2A* and *AF4*, respectively. Co-culture with stromal MS-5 cells was necessary for the successful expansion of transformed HSPCs, which strongly committed to the B-lineage. This presents common characteristics of acute lymphocytic leukemia and was not successful for long-term culture and expansion of genetically modified human KMT2A::AF4+ progenitors, as cell numbers declined rapidly after 7 weeks in co-culture. Nevertheless, using CRISPR-Cas9 to introduce the t(4;11) translocation in primary human HSPCs both reciprocal fusion genes, *KMT2A::AF4* and *AF4::KMT2A*, could be expressed. This technical improvement model *KMT2A::AF4* translocation in vitro is much closer compared to the retroviral fusion gene overexpression [42,43,44].

The second most common fusion partner AF9 similar to KMT2A plays a transcriptional role in embryonic development and hematopoiesis [45,46]. The t(9;11) translocation, resulting in the *KMT2A::AF9* fusion gene was shown to be capable of transforming human HSPCs in vitro by several groups. Mulloy and colleagues expressed the *KMT2A::AF9* cDNA using a retroviral vector construct in primary human umbilical cord blood CD34+ cells. The primary HSPCs were transformed and immortalized by KMT2A::AF9 expression. The KMT2A::AF9 long-term culture could be established without a feeder cell support [17].

Depending on the culture conditions such as cytokines, the KMT2A::AF9 expressing cells were either lineage committed or displayed a mixed-lineage phenotype [12,17]. Notably, those KMT2A::AF9 transformed HSPCs shared several characteristics with KMT2A::AF9 human primary patient samples and displayed limitless replicative potential, lineage commitment, and a very similar transcriptome. By comparing gene expression signatures of KMT2A::AF9+ cells, primary human AML samples, and RUNX1::RUNX1T1 or CBFβ MYH11 control cells, differently expressed gene sets were established. KMT2A::AF9+ cell culture samples strongly clustered with gene expression programs found in primary human AML samples and were even more closely related to the pediatric cohort rather than the adult acute myeloid leukemia [17]. More recently, genome editing was used as a non-viral-based method to introduce the *KMT2A::AF9* translocation into primary human HSPCs [39,47]. Here, transcription activator-like effector nucleases (TALENs) were utilized to introduce the t(9;11) translocation into CD34+ human cord blood cells. Similar to the CRISPR-Cas9 approach for *KMT2A::AF4*, the TALEN approach also generated both reciprocal forms, *KMT2A-AF9* and *AF9::KMT2A*. Despite the very low translocation frequency, KMT2A::AF9+ cells displayed monoclonal outgrowth, maintained their viability in long-term cultures, and kept proliferating and blocked maturation, resulting in an immature myelomonocytic phenotype [47].

The third most common *KMT2A* rearrangement in AML, t(11;19), results in the *KMT2A::ENL* fusion. The structure of *ENL* is highly similar to *AF9*. Physiologically, ENL acts as a histone modulator and transcriptional regulator, mostly through the highly conserved YEATS domain [48]. Precipitation and analysis of ENL-associated proteins revealed direct interaction with the histone H3K79 methyltransferase DOT1L, indicating a direct role of ENL in chromatin modification. Transcriptional elongation participation was shown by the interaction of ENL with subunits of pTEF-b and RNA Pol II CTD kinase. Further, ENL is associated with other frequent KMT2A partners such as AF4 [49]. Several research groups have demonstrated the establishment of human KMT2A::ENL+ long-term ex vivo cultures derived from primary HSPCs. Barabe and colleagues showed that retroviral expression of the *KMT2A::ENL* fusion gene itself is sufficient to initiate leukemogenic programs in primary human HSPCs to induce leukemia in vivo. When transplanting retrovirally transduced KMT2A::ENL+ human HSPCs directly into immunodeficient mice [50], recipient mice died within 20 weeks, showing pathologies also observed in human ALL patients. Secondary transplantation of BM from primary KMT2A::ENL injected mice led to the development of leukemia with the same phenotype as observed in the primary recipient, but with shorter latency. KMT2A::ENL+ HSPCs, cultured under myeloid conditions, induced B-ALL-like leukemia in vivo upon injection into immunodeficient mice. However, when injected after 50 days of culture, mice either developed a B-ALL, AML, or mixed phenotype leukemia. Injection of KMT2A::ENL+ cells after 70 days in ex vivo culture resulted strictly in an AML phenotype in vivo. No engraftment was observed when using KMT2A::ENL+ cells cultured for more than 90 days under myeloid conditions. These experiments demonstrate that the KMT2A::ENL fusion product can initiate both lymphoid and myeloid leukemogenesis. In addition, the observed lineage switch from ALL to AML is consistent with clinical data [51], where B-ALL is mostly found in pediatric patients with the *KMT2A::ENL* translocation and lineage switching is observed during the relapse [20]. Recently, we provided a detailed protocol for KMT2A::ENL-driven immortalization of human HSPCs in vitro. Using a retroviral approach to express the KMT2A::ENL fusion in human HSPCs, long-term ex vivo cultures of KMT2A::ENL+ monocytic progenitor cells could be established this way [52]. Further, several groups used TALEN and CRISPR/Cas9 genome-editing tools to induce the *KMT2A::ENL* translocation in primary human HSPCs. Buechele and colleagues introduced the *KMT2A::ENL* translocation into human cord blood HSPCs using a TALEN approach. *KMT2A::ENL* knock-in cells showed advanced survival and proliferation compared to control cells as well as higher clonogenic potential in CFU assay. However, engineered HSPCs were not fully transformed by the endogenous KMT2A::ENL expression, shown by proliferative exhaustion and enhanced differentiation when cultured > 4 months or further replated [53]. Similar results were obtained when using CRISPR/Cas9 to generate the t(11;19) translocation in human cord blood CD34+ HSPCs. Here KMT2A::ENL provided enhanced self-renewal and plating capacity but could not induce long-term proliferation in vitro. In contrast, transplantation of t(11;19) KMT2A::ENL+ HSPCs into immunodeficient mice led to long-term engraftment, enrichment of immature monocytic cells in the bone marrow as well as severe liver infiltration of myeloid cells, representing a malignant hematological phenotype. These findings indicate a critical role of the microenvironment and cellular context in modulating the transforming capacity of the KMT2A::ENL fusion [54] (Figure 1 and Figure 2).

### 2.2. CBF Rearrangements

The core binding factor (CBF) family consists of heterodimeric transcription factors with a non-DNA-binding beta chain (CBFB) and a DNA-binding alpha chain (CBFA). These proteins play an essential role in several stages of hematopoiesis [55] and are frequently involved in chromosomal translocations. The resulting fusion proteins disturb critical CBF-related functions especially cellular differentiation, thereby propelling leukemia [56]. One of the most commonly found and best-studied chromosomal translocation products associated with AML is the t(8;21) derived *RUNX1::RUNX1T1* (formerly known as *AML1-ETO*) fusion gene. Here, the breakpoints are located in intron 5 of *RUNX1*, also known as core-binding factor subunit alpha-2 (*CBFA2*), and intron 1 of *RUNX1T1*, resulting in a chimeric protein containing the *N*-terminus of RUNX1 and almost the entire RUNX1T1 protein. The RUNX1 part comprises the runt-homology domain responsible for DNA binding but lacks the transcriptional activation domain of the full-length RUNX1 [57]. The RUNX1T1 part on the other hand contains four nervy homology regions (NHR) and is a strong transcriptional modulator. Functionally, the NHR2 region orchestrates tetramer formation, either homo- or hetero-oligomerization of RUNX1T1 and different RUNX1T1 family members such as RUNX1T12 and MTGR1 [57]. The RUNX1::RUNX1T1 fusion protein interacts with a variety of epigenetic modulators such as histone deacetylases (HDACs) or DNA methyltransferases (DNMTs) and, together with co-factors, can act as a strong transcriptional repressor, affecting inter alia myeloid transcription, tumor suppressor genes, anti-apoptotic genes, genomic stability, and self-renewal capacities [58]. In particular, RUNX1T1’s NHR2 and NHR4 interactions with the co-repressors N-CoR/SMRT and mSin3a account for the transcriptional repressor properties [59]. Besides the repressor characteristic, RUNX1::RUNX1T1 can act as a transcriptional activator by binding the co-activator p300 via RUNX1T1’s NHR1 domain [60]. This interaction leads to p300-dependent acetylation of RUNX1::RUNX1T1, which is essential for both the self-renewal-fostering effects and the property to transactivate gene expression of the proto-oncogene *c-KIT* [60,61]. Retroviral RUNX1::RUNX1T1 overexpression in human CD34+ cells led to an oligo- or polyclonal outgrowth and establishment of long-term ex vivo cultures in a cytokine-dependent manner without feeder cell support required (Figure 1). Transduced cells showed high self-renewal capacity for over seven months, an immature phenotype with retained multi-lineage differentiation properties, as well as a normal karyotype [12,13,62]. When injected into immunocompromised NOD/SCID mice the RUNX1::RUNX1T1 transduced cells were able to engraft but did not elicit leukemia (Figure 2). Hence, overexpressing RUNX1::RUNX1T1 in human CD34+ as a single genetic element rather models a pre-leukemic phase of AML allowing for the investigation of additional hits to progress to leukemia [12,13]. In contrast, a truncated *RUNX1::RUNX1T1* isoform, generated via alternative splicing at exon 9, can elicit acute myeloid leukemia transformation in murine HSPCs [63]. The C-terminal truncated RUNX1::RUNX1T1 lacks RUNX1T1’s NHR3 and 4 domains rendering the protein a less potent repressor due to loss of interactions with NCoR/SMRT and HDACs [64]. This attenuated repressor function seems to be responsible for the enhanced leukemogenic potential as demonstrated by DeKelver and colleagues [65]. 

However, the in vitro proliferation rate of human CD34+ cells expressing truncated RUNX1::RUNX1T1 is not increased in comparison to full-length RUNX1::RUNX1T1. Instead, in co-culture studies full-length RUNX1::RUNX1T1 expressing cells outcompeted truncated RUNX1::RUNX1T1 expressing cells [64]. In general, both RUNX1T1’s characteristics to induce oligomerization and confer transcriptional repressor activity are essential to drive the ex vivo expansion of HSPCs. Deletion constructs of regions responsible for either function could not maintain proliferation concluding that oligomerization or transcriptional repression alone is not sufficient [62].

Inv(16) or t(16;16) represents a further chromosomal translocation involving a CBF transcription factor family member. Via the breakdown and rejoining of a segment of chromosome 16, a fusion of the *CBFB* gene to the *MYH11* gene occurs. With mouse models heterozygous for a *CBFβ::MYH11* knock-in into the endogenous *CBFβ* locus, it was demonstrated that the fusion protein operates as a repressor of CBF functions thereby disturbing the hematopoiesis [66]. Wunderlich and colleagues showed that retroviral overexpression of CBFβ::MYH11 in human hematopoietic CD34+ cells from healthy donors led to a sustained proliferation of transduced cells for up to seven months similar to RUNX1::RUNX1T1 overexpression. Yet, transduced cells displayed a strong myelomonocytic phenotype with a propensity to differentiate toward eosinophils in accordance with inv(16) leukemia which is associated with the AML subtype M4Eo [14].

Apart from chromosomal translocations and the generation of fusion genes, *RUNX1* mutations were found in myelodysplasia, de novo, and secondary AML. Such mutations are often associated with a loss of function and result in the expansion of common myeloid and granulocyte-macrophage progenitors. One of these mutants *RUNX1-S291fs300X* leads to a C-terminal truncation and was used in overexpression studies in human hematopoietic stem/progenitor cells by Gerritsen and colleagues [19]. Interestingly, this mutant alone was sufficient to increase human progenitor self-renewal and enhance long-term culture-initiating cell frequency by modulating specific gene expression patterns different from *RUNX1::RUNX1T1*. Upregulated genes in this context include *MEIS1* and *ERG*, which are associated with leukemogenesis.

### 2.3. NUP98::HOXA9 Rearrangement

Another group of AML-associated chromosomal rearrangements involves the nucleoporin 98 (*NUP98*) gene, fusing to over 28 different partner genes [67]. The NUP98 protein plays a major role in nuclear import and export as part of the nuclear pore complex. However, it was demonstrated that NUP98 participates in gene transcription with its amino-terminal transactivation domain [67]. Following chromosomal translocation, the fusion partners can be classified into either homeodomain-containing proteins or non-homeodomain proteins. The first category covers a variety of homeobox (HOX) genes with the fusion protein NUP98::HOXA9 generated by a t(7;11) translocation being the most common and best-studied fusion. Similarly to the other rearrangements, the t(7;11) translocation results from the fusion of the *N*-terminus of *NUP98* and the C-terminal of *HOXA9*. Using the human myeloid cell line K562, it has been found that NUP98::HOXA9 acts as a transcriptional activator and is associated with enhanced proliferation and blocked differentiation [68]. In general, NUP98 fusions are associated with high expression of HOXA7, HOXA9, HOXA10, and the co-factor MEIS1 thereby deregulating normal hematopoiesis. Takeda and colleagues studied the effects of retrovirally overexpressed *NUP98::HOXA9* in primary human hematopoietic progenitor cells. Hindered myeloid and erythroid differentiation and enhanced self-renewal of immature cells were observed when NUP98::HOXA9+ progenitor cells were seeded on semisolid media. In liquid cultures without feeder cell support, transduced CD34+ cells initially showed growth inhibition, most likely due to IFNβ1 up-regulation, followed by long-term proliferation of immature cells for nearly eight weeks, before cell numbers declined [69]. Similarly, Chung and colleagues observed an in vitro proliferative advantage of *NUP98::HOXA9* expressing human CD34+ cells in both stromal co-cultures and without feeder cell support in cytokine-stimulated liquid cultures. These cells exhibited a myeloblast/promyelocytic phenotype with an up to 10-fold greater expansion capacity compared to empty vector-transduced control cells. The committed progenitor cells were able to form secondary and tertiary myeloid colonies highlighting their self-renewal potential. Moreover, when injected in NOD/SCID, NOD/SCID β_2_M^null^, and NOD/SCID γ2^null^ mice [50], engraftment was obtained with untransduced CD34+ cells, NUP98::HOXA9-expressing and empty vector-transduced cells. Within 5 to 6 weeks a significant increase in NUP98::HOXA9 positive cells was observed emphasizing their proliferative advantage in vivo. These cells displayed a primitive myelomonocytic blast morphology [18]. Similar in vivo results were obtained with the fusion protein NUP98::HOXA10HD, where the *N*-terminal part of *NUP98* is fused only to the homeodomain (HD) of *HOXA10*. However, in vitro data revealed that NUP98::HOXA10HD expression in human CD34+ cells did not result in an enhanced proliferation and self-renewal [70].

### 2.4. PML::RARA Rearrangement

Acute promyelocytic leukemia (APL), a subtype of AML, is defined by the translocation t(15;17) resulting in the formation of the PML::RARA fusion. Over 98% of APL patients harbor this specific translocation [71]. Here the nuclear retinoic acid receptor *RARA* fuses nearly entirely to *PML (*promyelocytic leukemia protein). RARα as a member of the retinoic acid receptor family (RARs) acts as a nuclear transcription factor that forms heterodimers with retinoid X receptors (RXR). In the absence of a ligand DNA-bound RXR/RARA acts as a transcriptional repressor by recruiting specific co-factors. Upon retinoic acid (RA) ligand binding, the complex undergoes conformational changes and can subsequently recruit coactivators, resulting in transcriptional activation of RARα target genes [72,73]. Hence RARα influences both proliferation and differentiation as well as survival and apoptosis [74]. PML is described as a tumor suppressor, mainly acting via the associated PML-nuclear bodies (PML-NBs) influencing transcriptional regulation, e.g., activation of p53 signaling, apoptosis, senescence, DNA damage response, and viral defense mechanisms via its different isoforms [75].

The fusion product PML::RARA interferes both with the RARA signaling as well as the PML-NBs. First, PML::RARA is a dominant negative toward RXR/RARA, inhibiting transcriptional activation of target genes resulting in blocked differentiation and enhanced self-renewal of promyelocytes. Second, PML::RARA modulates PML-NBs in a way that inhibits apoptosis, senescence, and DNA damage response pathways [76].

APL cells are relatively differentiated [77], which makes it unclear whether the oncogenic effect occurs on the stem cell level or further down in the hierarchy. There are indications that primitive HSPCs are not involved in the onset of APL [78]. Nevertheless, several groups retrovirally transduced human HSPCs with PML::RARA cDNA to investigate the effects of this translocation in vitro. Retrovirally transduced PML::RARA could establish a leukemic promyelocytic phenotype in peripheral blood HPC/HSC through the reprogramming towards promyelocytic differentiation followed by inhibition of further differentiation. But PML::RARA+ cells quickly stopped proliferating and long-term culture could not be established [79,80].

In sharp contrast to this, murine HSPCs retrovirally transduced with PML::RARA continued proliferating, demonstrated by high colony-forming ability with several rounds of replating [81], highlighting fundamental differences between murine and human cells.

Primary human APL samples are very difficult to engraft in immunodeficient mice, however, there are xenograft APL models in which PML::RARA was introduced into human cord blood HSPCs and transplanted into immunodeficient, humanized NOG mice [82,83]. Before transplantation, PML::RARA+ cells showed several features of APL in vitro, such as the disruption of PML-NBs and induction of promyelocytic differentiation. Further PML::RARA dramatically reduced the colony-forming capacity. In vivo PML::RARA+ cells engrafted and induced APL, which reflected the phenotype of the human disease both genetically and functionally [83].

## 3. “Two-Hit” Models for Human CD34+ Ex Vivo Progenitor Cell Expansion

During AML development the acquisition of multiple mutations induces full-blown leukemia. This stepwise development can be modeled in CD34+ ex vivo cultures by introducing defined oncogenes. Rizo and colleagues showed that oncogenic cooperativity between the BCR::ABL fusion and the polycomb gene *BMI1* promotes long-term ex vivo proliferation and self-renewal capacity of transduced human CD34+ HSPC cells [16]. BMI1 is essential for normal HSPC maintenance and self-renewal and is downregulated during differentiation. Through its association with the Polycomb repressor complex 1, BMI1 acts as a transcriptional repressor through epigenetic modifications and regulates cell cycle progression, apoptosis, and senescence [84,85]. Enhanced expression of *BMI1* was detected in a variety of cancer entities, including hematological malignancies such as CML and AML, and serves both as a biomarker and prognostic marker [86]. To show oncogenic cooperation between BMI1 and BCR::ABL, Rizo and colleagues retrovirally transduced human cord blood CD34+ cells with either *BCR::ABL*, *BMI1*, or both, and cultured the obtained cells under stromal cell support and a myeloid cytokine mix. After five weeks of culture, the cells were tested for replating capacity on stromal cells. BMI1+ cells could be replated once, but during the third replating, no self-renewal and expansion were observed. BCR::ABL+ cells on the other hand could not form any CFUs after replating. In contrast, double transduced BMI1+/BCR::ABL+ cells showed advanced growth throughout several rounds of replating and maintained their proliferative signaling for over 20 weeks of permanent ex vivo culture.

Morphologically, BMI1/BCR::ABL double positive long-term cultures showed a blast-like, myeloid precursor phenotype, consistent with their ability to establish long-term ex vivo cultures. In line with the previously described observations, the retroviral introduction of *BMI1* into BCR::ABL+ primary CML patient samples enhanced self-renewal and proliferation in ex vivo cultures compared to untransduced CML cells. Finally, BCR::ABL+, BMI1+, and BCR::ABL+/BMI1+ CD34+ cells were transplanted into NOD/SCID mice to test for leukemia induction in vivo. BCR::ABL+, as well as BMI1+ cells alone, were not able to introduce leukemia within 25 weeks post-transplantation, whereas 50% of BCR::ABL+/BMI1+ mice succumbed to leukemia within weeks. Strong proliferative advantage of BCR::ABL+/BMI1+ cells was observed as the percentage of double-positive cells highly increased in the bone marrow and peripheral blood cells. Secondary transplantation showed robust engraftment and even shorter latency with a lymphoblastic phenotype [16,87].

Even though the RUNX1::RUNX1T1 fusion was shown to be sufficient to induce ex vivo expansion of transduced human CD34+ cells [13] the transforming capacity of RUNX1::RUNX1T1 can be dramatically enhanced by additional activation of the c-KIT receptor tyrosine kinase. C-KIT is widely expressed among several tissues but plays a particularly important role in hematopoiesis by controlling HSPC maintenance, proliferation, and differentiation. Upon SCF ligand binding c-KIT activates numerous downstream pathways such as the MAPK/ERK or the PI-3 kinase pathway, both associated with proliferation and survival [88]. Activated *c-KIT* mutations are frequently found in AML patients leading to enhanced oncogenic signaling [89]. Recently, we demonstrated that full-length or truncated RUNX1::RUNX1T1 together with the constitutively active c-KIT mutant N822K, triggered rapid outgrowth of transduced human peripheral blood CD34+ HSPCs, and established extensive long-term proliferating ex vivo cultures without requirement of feeder cell support. Whereas RUNX1::RUNX1T1 together with c-KIT(N822K) showed highly enhanced outgrowth potential, this could be not reproduced with RUNX1::RUNX1T1 or c-KIT(N822K) alone. Notably, co-expression of RUNX1::RUNX1T1 with wildtype c-KIT also could not establish enhanced outgrowth of human precursor cells. Hematopoietic progenitor cells expressing both, RUNX1::RUNX1T1 and c-KIT(N822K), showed growth advantage and enhanced differentiation blockage as well as overgrowth of RUNX1::RUNX1T1 single-positive cells in vitro. Morphologically, RUNX1::RUNX1T1 + c-KIT(N822K) cells displayed a more primitive phenotype compared to single transduced cells. Still, both populations were able to form colonies when plated on methylcellulose several months post-transduction. However, in serial replatings, only RUNX1::RUNX1T1 + c-KIT(N822K) cells continued proliferating [90]. In line with these observations, Chin and colleagues confirmed the oncogenic cooperation of RUNX1::RUNX1T1 with c-KIT(N822K) in human cord blood CD34+ cells. In their model, RUNX1::RUNX1T1 + c-KIT(N822K) also displayed growth advantage, enhanced expansion ability, and clonogenic potential compared to RUNX1::RUNX1T1 alone [91]. As mentioned, RUNX1::RUNX1T1 alone was not able to induce leukemogenesis in mice, prompting the question of additional co-activators necessary to induce leukemia in mice [92]. However, a truncated version of RUNX1::RUNX1T1 was reported to directly induce leukemia in mice [93,94]. Further, co-expression of full-length *RUNX1::RUNX1T1* with activated *c-KIT* mutations efficiently triggered leukemia in vivo upon transplantation of transduced murine HSPCs into mice [95,96]. However, dramatically different outcomes were observed when transplanting primary human HSPCs into NSG mice. Even though co-expression of RUNX1::RUNX1T1 with c-KIT or other co-mutations led to enhanced in vitro growth of primary human cord blood cells, no induction of leukemogenesis was observed in xenograft models (Figure 2), pointing out fundamental differences between the human and murine systems [90,97]. It is therefore suggested that in the human context, more than “two hits” are necessary for full progenitor cell transformation, making the human system much more robust against oncogenic stress than the murine counterpart [12,58,98,99].

## 4. Synopsis

Ex vivo expansion of human CD34+ hematopoietic stem/progenitor cells is a hallmark of distinct leukemic transcription factors found in acute myeloid leukemia. Normally, stem/progenitor cells require the full support of stem cell niche factors for proper self-renewal. Cytokines alone cannot prohibit cell differentiation along the lineages. However, expression of AML-associated fusion genes in combination with cytokines allows for long-term ex vivo expansion making these cell models highly attractive to study the biology of oncogenes, as well as therapeutic avenues to block AML oncogene-dependent cell proliferation (Figure 3). As discussed above, primary human blood progenitor cells show many advantages compared to murine stem cells, especially working in the same host organism as the disease occurs. Even though primary human HSPCs are not as easily available as their murine counterpart, there are still different options for obtaining human cells. The two main resources of human cells are either remaining cells obtained from consent patient donations or samples from healthy individuals commercially available. Besides the many advantages of primary human ex vivo AML models, there are nevertheless limitations. Ex vivo cultures cannot model the complexity of the tumor microenvironment or the heterogeneity of many AML subtypes [100]. Further, the specific culture conditions can affect the phenotype and cannot represent complex physiological conditions such as oxygen levels [101]. Xenograft models, in which human leukemia cells are transplanted into immunodeficient mice, aim to overcome some of those limitations but still lack certain important features of AML. As leukemic cells are transplanted into immunodeficient mice to avoid rejection of the injected cells, there is no interaction between leukemia cells and the immune system, which can affect the disease progression and drug response [102,103,104]. Also, species-specific differences in the microenvironment, metabolism, and signaling pathways can lead to biased results [35,105]. Taken together, no artificial model system can reflect the complexity of the human organism, but ex vivo models are a very valuable tool to study human leukemic cells and should be used additionally or complementary to cell lines and in vivo models.

## Figures and Tables

**Figure 1 cells-13-00078-f001:**
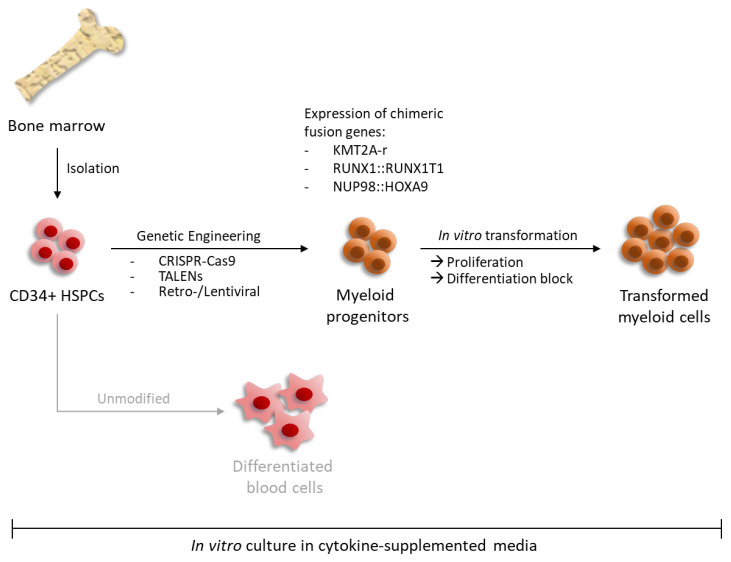
Modeling oncogenic transformation of myeloid blood progenitors in ex vivo cultures via genetic engineering. Human CD34+ HSPCs isolated from bone marrow or cord blood of healthy donors can be genetically manipulated to mimic leukemic transformation. Expression of leukemia-associated fusion genes can be induced either via retro- or lentiviral integration or by CRISPR-Cas9/TALENs-mediated rearrangements. In cytokine-supplemented media, continuous expression of these chimeric genes leads to enhanced proliferation and concomitant differentiation block. Unmodified CD34+ HSPCs are not able to continuously proliferate in vitro resulting in terminal differentiation.

**Figure 2 cells-13-00078-f002:**
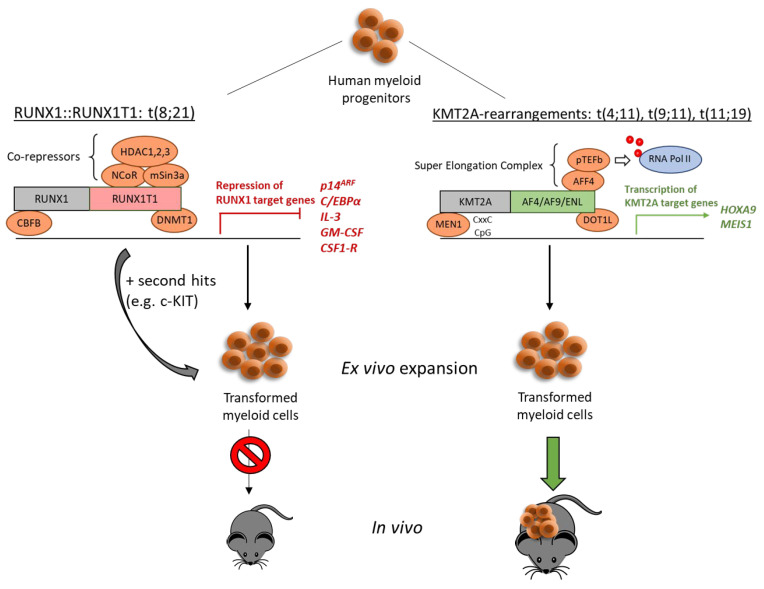
Utilizing oncogenes for leukemic transformation of human myeloid progenitor cells ex vivo and in vivo. Transduction with both either KMT2A-r or RUNX1::RUNX1T1 leads to ex vivo transformation and expansion. RUNX1::RUNX1T1 primarily acts as a transcriptional repressor, while KMT2A-rearrangement-derived fusion proteins function as activators. Whereas KMT2A fusion proteins are strong independent leukemic drivers, the transforming capacity of RUNX1::RUNX1T1 must be amplified by further mutations such as c-KIT. Transplantation of transformed human cells into immune-deficient mice only leads to leukemia in the case of KMT2A-r, but not RUNX1::RUNX1T1 both with and without co-mutations.

**Figure 3 cells-13-00078-f003:**
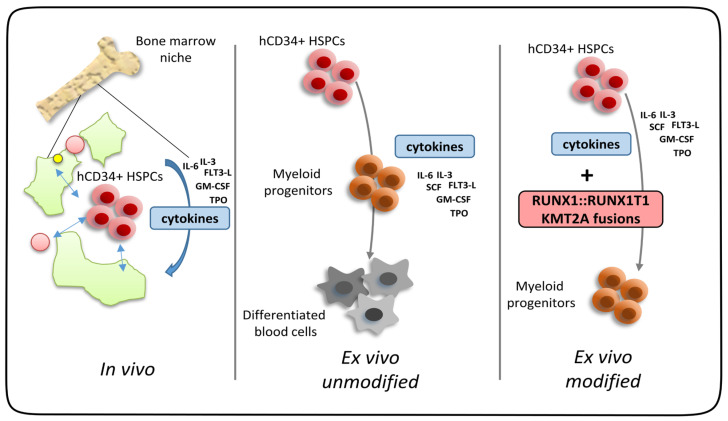
AML-associated fusion genes such as RUNX1::RUNX1T1 and KMT2A-fusions can substitute for hematopoietic stem cell niche factors to allow for ex vivo progenitor cell expansion under liquid culture conditions. Detached from the bone marrow niche unmodified CD34+ cells rapidly differentiate ex vivo even in the presence of proliferation-favoring cytokines. Together with cytokines, distinct leukemic transcription factors induce niche-independent expansion of hematopoietic progenitor cells in ex vivo cultures.

**Table 1 cells-13-00078-t001:** Classification of acute myeloid leukemia with defining genetic abnormalities. Adopted from [9,10,11].

Genetic Abnormality	Resulting Fusion Gene/Rearrangement
t(8;21)	*RUNX1::RUNX1T1*
t(16;16) or inv(16)	*CBFB::MYH11*
t(6;9)	*DEK::NUP214*
t(3;3) or inv(3)	*RPN1::EVI1*
t(1;22)	*RBM15::MRTFA*
t(9;22)	*BCR::ABL1*
t(15;17)	*PML::RARA*
t(11q23)	*KMT2A* (*MLL1, MLL*)-r
t(3q26)	*MECOM*-r
t(11p15)	*NUP98*-r
biallelic mutations of the *CEBPA* gene	-
with other defined genetic alterations	-

**Table 2 cells-13-00078-t002:** Overview of oncogenes and culture conditions for human ex vivo CD34+ liquid culture models. Genes that alone or in combination can induce human CD34+ cell expansion in ex vivo liquid cultures. For each mutated gene, the required cytokine combination and resulting cellular phenotype are listed.

Phenotype	Gene	Culture Conditions	References
Myeloid, CD34+ subpopulation and mature cells	*RUNX1::RUNX1T1*	SCF, IL-3, IL-6, GM-CSF, FLT3-L, erythropoietin	[12,13]
	*CBFβ::MYH11*	SCF, IL-3, IL-6, FLT3-L, thrombopoietin	[14]
	*KMT2A::AF4*	IL-3, SCF, FLT3-L	[15]
	*BCR::ABL* + *BMI1*	IL-3, IL-6, G-CSF, SCF, FLT3-L MS5 stromal feeder cells	[16]
Myeloid progenitors, no remaining CD34+ subpopulation	*KMT2A::AF9*	SCF, IL-3, IL-6, FLT3-L and TPO	[17]
	*NUP98::HOXA9*	SCF, FLT3-L, thrombopoietin	[18]
Immature granulocyte-macrophage progenitor-like cells	*RUNX1* mutations e.g., *RUNX1-S291fs300X*	IL-3, IL-6, G-CSF, SCF, FLT3-L, erythropoietin	[19]
Both myeloid and lymphoid progenitors	*KMT2A::ENL*	IL-3, SCF, FLT3-L, IL-7	[20]

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
