# Peer review of "Deciphering Acute Myeloid Leukemia Associated Transcription Factors in Human Primary CD34+ Hematopoietic Stem/Progenitor Cells"

_cells, 2023, doi:10.3390/cells13010078_

Round 1

Reviewer 1 Report

Comments and Suggestions for Authors

In their review, Kreissig et al addresses the issue of genetic abnormalities in patients with myeloid neoplasms reviewing the role of specific chimeric fusion genes on expansion of myeloid progenitors up to AML development.

Major issues

- AML section: the authors insert in this section a paragraph stating “The first chromosomal rearrangement detected in a patient with chronic myeloid leukemia (CML) was the so-called Philadelphia Chromosome”: it is not clear why they report this finding for AML where it is an absolute rarity. I would suggest removing this paragraph from this section.

- AML section: the authors quote WHO classification, I would suggest quoting ICC also (Arber Blood 2022)

Minor issues

- The authors should use the updated nomenclature for gene rearrangemnents (i.e., BCR::ABL instead of BCR-ABL).

Author Response

Thank you very much for taking the time to review this manuscript and the constructive criticism. Please find the detailed responses below.

Major issues

- AML section: the authors insert in this section a paragraph stating “The first chromosomal rearrangement detected in a patient with chronic myeloid leukemia (CML) was the so-called Philadelphia Chromosome”: it is not clear why they report this finding for AML where it is an absolute rarity. I would suggest removing this paragraph from this section.

This paragraph was removed/rewritten as requested (line 52-53).

- AML section: the authors quote WHO classification, I would suggest quoting ICC also (Arber Blood 2022)

The ICC reference was included besides the WHO classification (line 55-61).

Minor issues

- The authors should use the updated nomenclature for gene rearrangemnents (i.e., BCR::ABL instead of BCR-ABL).

The nomenclature has been updated as requested.

Reviewer 2 Report

Comments and Suggestions for Authors

The underlying argument for this review article is that ex vivo expansion of human CD34+ cells engineered to express leukemia-specific fusion genes more faithfully model the human disease because of differences between mouse and human. Although they do provide limited examples where mouse models and CD34+ models have supported different conclusions, they do not acknowledge any of the limitations of ex vivo culturing or xenograft mouse models, which are just as likely to contribute to the different conclusions as differences between mouse and human hematopoietic cells. A more balanced discussion of how recent advances in CD34+ engineered leukemia models can complement existing mouse models and patient-derived xenograft models would be more helpful, particularly to those new to the field.

Minor comment: HUGO nomenclature now recommends using :: to indicate fusion genes, ie (BCR::ABL) and using KMT2A rather than MLL and RUNX1::RUNX1T1 instead of RUNX1::ETO.

Line 34-35 states that AML is the most common leukemia in adults. That is incorrect. CLL is the most common leukemia in adults. AML is the most common acute leukemia in adults.

Comments on the Quality of English Language

Minor proofreading is required.

Author Response

Thank you very much for taking the time to review this manuscript and the constructive criticism. Please find the detailed responses below.

The underlying argument for this review article is that ex vivo expansion of human CD34+ cells engineered to express leukemia-specific fusion genes more faithfully model the human disease because of differences between mouse and human. Although they do provide limited examples where mouse models and CD34+ models have supported different conclusions, they do not acknowledge any of the limitations of ex vivo culturing or xenograft mouse models, which are just as likely to contribute to the different conclusions as differences between mouse and human hematopoietic cells. A more balanced discussion of how recent advances in CD34+ engineered leukemia models can complement existing mouse models and patient-derived xenograft models would be more helpful, particularly to those new to the field.

We agree to this point. The discussion has been updated to be more balanced (line 482 -496).

Minor comment: HUGO nomenclature now recommends using :: to indicate fusion genes, ie (BCR::ABL) and using KMT2A rather than MLL and RUNX1::RUNX1T1 instead of RUNX1::ETO.

The nomenclature has been updated as requested.

Line 34-35 states that AML is the most common leukemia in adults. That is incorrect. CLL is the most common leukemia in adults. AML is the most common acute leukemia in adults.

This sentence has been rewritten accordingly, stating AML is the most common type of acute leukemia.

Reviewer 3 Report

Comments and Suggestions for Authors

This is an excellent review of development of human AML models via introduction of AML oncogenes into human HPSC.  I suggest adding information regarding PML-RARA models, e.g. PMC4219701

Author Response

Thank you very much for taking the time to review this manuscript and the constructive criticism. Please find the detailed responses below.

As suggested an additional paragraph on PML::RARA has been added and in vitro as well as in vivo models were discussed (line 332-370).

Reviewer 4 Report

Comments and Suggestions for Authors

Authors described how the various oncogenic transcription factors (mainly chimeric transscription factors resulting from chromosomal translocations) contributed to leukemic transformation of human CD34+ human hematopoietic stem and progenitor cells (HSPCs). Zinc finger nucleases, transcription activator-like effector nucleases (TALENs), and clustered regularly interspaced short palindromic repeats (CRISPR)/Cas systems were used for targeted genome editing. I think that some references to these methods should be used, for example Carroll, D. Genome engineering with targetable nucleases. Annu Rev Biochem 2014, 83, 409-439.; Ferrari, S.; Vavassori, V., Canarutto, D., Jacob, A.; Castiello, M.C.; Javed, A.O., Genovese, P. Gene editing of hematopoietic stem cells: hopes and hurdles toward clinical translation. Front Genome Edit 2021, 3, 618378. Thesis of Dr. Shan Lin, 2016 University of Cincinnati -Modeling and analysis of acute leukemia using hematopoietic stem and progenitor cells. https://etd.ohiolink.edu/acprod/odb_etd/ws/send_file/send?accession=ucin14810321447804128disposition=inline     could be also mentioned.

It is possible that some readers of this journal do not know what is the NSG mouse (NOD scid gamma mouse)-a brand of immunodeficient laboratory mice. Reference: Chen, J.; Liao, S.; Xiao, Z.; Pan, Q.; Wang, X.; Shen, K.; Wang, S.; Yang, L.; Guo, F.; Liu, H.F.; Pan, Q. The development and improvement of immunodeficient mice and humanized immune system mouse models. Front Immunol 2022, 13, 1007579. could be also involved.   

Author Response

Thank you very much for taking the time to review this manuscript and the constructive criticism. Please find the detailed responses below.

We thank the reviewer for this point! The additional references were added regarding methodology.   

Round 2

Reviewer 2 Report

Comments and Suggestions for Authors

The authors have adequately addressed my concerns.